# Influence of shape resonances on the angular dependence of molecular photoionization delays

F. Holzmeier [1,2,5 ✉], J. Joseph [1], J. C. Houver[1], M. Lebech[3], D. Dowek[1] & R. R. Lucchese [4 ✉]

Characterizing time delays in molecular photoionization as a function of the ejected electron emission direction relative to the orientation of the molecule and the light polarization axis provides unprecedented insights into the attosecond dynamics induced by extreme ultraviolet or X-ray one-photon absorption, including the role of electronic correlation and continuum resonant states. Here, we report completely resolved experimental and computational angular dependence of single-photon ionization delays in NO molecules across a shape resonance, relying on synchrotron radiation and time-independent ab initio calculations. The angle-dependent time delay variations of few hundreds of attoseconds, resulting from the interference of the resonant and non-resonant contributions to the dynamics of the ejected electron, are well described using a multichannel Fano model where the time delay of the resonant component is angle-independent. Comparing these results with the same resonance computed in e-$NO^+$ scattering highlights the connection of photoionization delays with Wigner scattering time delays.

[1] Université Paris-Saclay, CNRS, Institut des Sciences Moléculaires d'Orsay, 91405 Orsay, France. [2] Université Paris-Saclay, Synchrotron SOLEIL, 91190 Saint Aubin, France. [3] Niels Bohr Institute, University of Copenhagen, Copenhagen, Denmark. [4] Lawrence Berkeley National Laboratory, Berkeley, CA 94720, USA. [5] Present address: imec, 3001 Leuven, Belgium. ✉email: fabian.holzmeier@imec.be; rlucchese@lbl.gov

The interaction of a single XUV photon with a molecule induces ultrafast photoionization dynamics, in which electronic coherences and many-body interactions, including couplings between electron and nuclear motions, are pivotal[1,2]. In addition, resonances, i.e., quasi-bound states in the ionization continuum, often have a significant impact on the dynamics due to enhanced cross-sections or trapping of the outgoing electron. Characterizing the photoionization dynamics in molecules at its natural timescale stands therefore as a vital research field at the forefront of attosecond science[3,4].

In recent years photoionization time delays were characterized in real-time pioneering experiments where one-photon ionization induced by an XUV attosecond pulse is probed in a delayed femtosecond IR field using RABBITT[5] or streaking[6] techniques. For atomic targets, one-photon ionization delays could be retrieved from the time-dependence of the oscillatory two-photon ionization signal demonstrating, e.g., relative photoionization time delays of few tens of attoseconds between electron ejection from $ns$ and $np$ electronic orbitals in Ne and Ar[7,8] or addressing the characterization of autoionizing Fano resonances in the ionization continuum[9–11]. More insight into such ionization dynamics described by two-photon transition matrix elements was then demonstrated in angular resolved studies in the laboratory frame[12–18].

Measuring photoionization delays in molecular targets using similar techniques is more challenging. First, the higher density of electronic states and their intrinsic energy width result in congested electron energy spectra when ionization is induced by the broadband XUV pulse and probed by the IR fundamental field. Using RABBITT attosecond interferometry where the XUV spectrum consists of a harmonic frequency comb allowing for spectral resolution, this could be circumvented to some extent, providing photoionization delays between different valence ionic states of randomly oriented small molecular targets[19–23]. It was proven e.g. that shape resonances, caused by the transient trapping of the photoelectron in an angular momentum barrier[24] structuring the continuum of ionized states, can lead to positive two-photon ionization delays larger than one hundred attoseconds[20–23]. Meanwhile, it was recognized that the latter observables do not give access to the one-photon ionization dynamics, due to the complexity inherent to the non-spherical symmetry of the molecular objects[20,25]. Addressing the angular dependence of the photoionization dynamics in molecules as well adds complexity since, even for the simplest diatomic molecules, it depends on both the orientation of the molecule relative to the light quantization axis $\hat{\Omega}$, and the emission direction of the photoelectron in the molecular frame (MF) $\hat{k} \equiv (\theta_k, \phi_k)$.

The one-photon ionization-time delay $\tau(\hat{k}, \hat{\Omega}, E)$, where $E$ is the electron energy, is defined as the energy derivative of the phase $\eta(\hat{k}, \hat{\Omega}, E)$ of the complex-valued angle-dependent photoionization dipole amplitude (PDA) $D(\hat{k}, \hat{\Omega}, E)$,

$$\tau(\hat{k}, \hat{\Omega}, E) = \frac{d\eta(\hat{k}, \hat{\Omega}, E)}{dE} \quad (1)$$

which is equivalent to the definition of the group delay for a wavepacket. Energy and MF angle-resolved attosecond photoionization delays for specific orientations of a diatomic molecule parallel or perpendicular relative to the axis of linearly polarized light were theoretically predicted for single-photon valence ionization of, e.g., $N_2$ and $CO$[26], highlighting indeed a strong anisotropy of the ionization dynamics. The rich angular patterns were attributed to the intrinsic anisotropy of the molecular potential featuring the electronic structure of the ionized molecular states and accounted for by the interference of the many partial waves with quantum numbers $(l, m)$ building up the electron wave packet in the continuum.

Initial experimental evidence of an orientation-dependent anisotropy of the dynamics was demonstrated in RABBITT studies addressing non-resonant inner valence ionization of $CO$ through the determination of stereo Wigner time delays, i.e., the difference in time delay for emission within a $2\pi$ solid angle towards the C and O ends of the molecule[27], or dissociative ionization of $H_2$ in presence of autoionizing resonances and coupling between electron and nuclear motions[28].

In the present study, experimental and computed photoionization time delays with complete angular resolution in the molecular frame are reported and analyzed in terms of the influence of a well-identified $\sigma^*$ shape resonance[29] in inner-valence ionization of the NO molecule. Experimental ionization delays are obtained from single-photon ionization measurements of molecular frame photoelectron angular distributions (MFPADs)[30,31] using synchrotron radiation at a series of well-resolved photon energies. In order to interpret the observed energy and MF-angle resolved dynamics, varying within a few hundreds of attoseconds, a multichannel Fano formalism is presented, from which we obtain separately the contributions of resonant and non-resonant photoionization amplitudes that coherently add to yield the total PDA. The results demonstrate that the angular dependence of the MF-resolved time delays is a signature of the interference between the resonant and non-resonant components of the PDA, where the ionization delay of the resonant component is angle-independent. Beyond the increase by $\pi$ of the scattering phase shift across a resonance[32], we show that in resonant molecular photoionization the PDA phases for different emission directions can change by values ranging from 0 to $2\pi$, giving rise to diverse angle-dependent time delay energy profiles. These behaviors can be most conveniently predicted by considering the curve formed by the PDA in the complex plane when plotted at different energies, for selected emission angles in the molecular frame. Finally, a comparison with the same resonance computed in e-$NO^+$ scattering gives more insight into the connection between Wigner scattering time delays and photoionization time delays.

## Results

**Angle-resolved photoionization time delays across the NO shape resonance.** MF-angle resolved photoionization delays for single-photon ionization of the NO molecule into the $NO^+(c\,^3\Pi)$ inner valence ionic state are extracted from the MFPADs measured using circularly polarized synchrotron radiation[33], for a series of eleven well-resolved photon energies probing the 23.25–38.75 eV range at equal steps of 1.55 eV (see the "Methods" section). This region covers the well-identified shape resonance assigned to ionization into the $NO^+(c\,^3\Pi)$ for the parallel transition, featured by a light polarization parallel to the molecular axis[34], with a maximum of the cross-section occurring around 31 eV photon energy ($\sim$9 eV photoelectron energy). The MFPAD, i.e., the photoemission probability along $\hat{k} = (\theta_k, \phi_k)$ in the molecular frame for any orientation of the molecule relative to the light quantization axis $\hat{\Omega}$, is proportional to the absolute square of the complex-valued photoionization dipole amplitude $D(\hat{k}, \hat{\Omega}, E)$. The latter is written as a coherent superposition of the partial wave dipole matrix elements (DMEs) $D_{l,m,\mu}$ with an angular dependence described by the spherical harmonics $Y_{l,m}(\hat{k})$, as

$$D(\hat{k}, \hat{\Omega}, E) = \sum_{\mu}\sum_{l,m} D_{l,m,\mu}(E) Y_{l,m}(\hat{k}) R^{(1)}_{\mu,h}(\hat{\Omega}), \quad (2)$$

where $R^{(1)}_{\mu,h}(\hat{\Omega})$ is the matrix rotating the molecular frame into the field frame defined by the Euler angles $\hat{\Omega} \equiv (\gamma, \chi, \beta)$, and $\mu$ is the

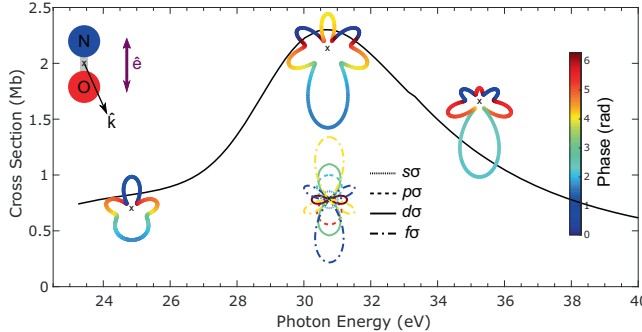

**Fig. 1 Resolving the complex-valued photoionization amplitudes in the molecular frame across a shape resonance.** Energy dependence of the complex-valued photoionization dipole amplitudes for the ionization of NO leading to the $(4\sigma)^{-1}$ $c$ $^{3}\Pi$ state of $NO^{+}$ plotted as a function of the emission direction $\hat{k}$ in the molecular frame for molecules oriented parallel to linearly polarized light in the region of the NO $4\sigma \rightarrow k\sigma^{*}$ shape resonance. The computed magnitudes and phases are represented by the radius vector and the color of the line plots, respectively. The energy derivative of the PDA phases gives access to photoionization time delays. For photon energy of 31 eV at the peak of the resonance, the PDAs are dominated by the coherent superposition of partial waves up to $l = 3$ as shown.

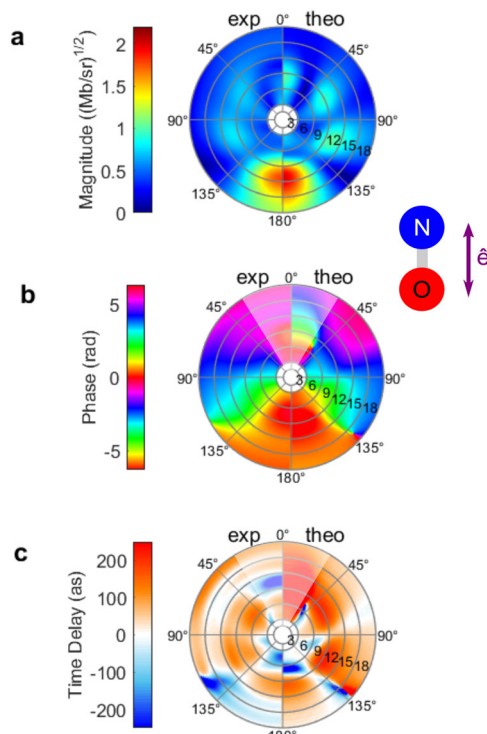

**Fig. 2 Magnitude and phase of PDAs and photoionization time delays.** Polar plots of **a** the PDA magnitudes, **b** PDA phases, and **c** photoionization time delays. The left (right) half of each map corresponds to the experimental (computed) data. The radial distance from the center of the maps gives the photoelectron kinetic energy $E$ in eV. The polar angle $\theta_k$ is relative to the NO molecular axis, with ($\theta_k = 0°$) corresponding to the direction of the N atom. In this and all subsequent figures, magnitudes have been multiplied by $\sqrt{4\pi^2 h\nu/c}$ so that their absolute squares yield the differential cross-section in Mbarn/steradian. See text and Fig. S4 (Supplementary Note 7) for corresponding uncertainties.

projected angular momentum of the photon on the MF axis, with $h$ indicating the light polarization (see Supplementary Note 3). The partial wave expansion of the photoionization dipole amplitude in Eq. (2) is at the core of the derivation of the ionization delays defined in Eq. (1). This approach is sketched in Fig. 1 for an orientation of the molecule parallel to the polarization axis.

In the experiment, the first major step consists of extracting the complex-valued $D_{l,m,\mu}$ DMEs for the individual $(l, m)$ partial-waves from the measured MFPADs, their magnitudes $d_{lm}$ and relative phases $\widetilde{\eta}_{lm}$, for each measured photon/electron energy[33,35] across the shape resonance. A finer sampling of the region of the resonance is achieved using a spline fit of the measured $D_{l,m,\mu}$ magnitudes and relative phases (see Supplementary Note 4). Equation (2) then provides the corresponding PDAs for any MF emission direction $\hat{k}$ and orientation of the molecule $\hat{\Omega}$, their magnitude, and phase $\eta(\hat{k}, \hat{\Omega}, E)$. In order to establish coherence between the phases of the PDAs for different energies, the relative phases of the partial wave DMEs extracted for each energy were based on a common reference: in the presence of resonance, it is advantageous to choose a reference DME $D_{l,m,\mu}$ that is not coupled to the resonance (see Supplementary Notes 4 and 5). Finally, photoionization time delays are obtained as the energy derivative of the PDA phases according to Eq. (1).

In the following, we focus on the PDAs with NO oriented parallel to the axis of linearly polarized light, which highlights the interference between the shape resonance and the non-resonant direct ionization channel in the parallel transition. Each PDA is thus characterized by the single polar angle $\theta_k$ due to the cylindrical symmetry of MF photoemission in that case, and it involves solely $(l, 0) \equiv (l, \sigma)$ partial waves, with here $l_{max} = 3$. Experimental data are compared with time-independent calculations using the multi-channel Schwinger configuration interaction method (see the "Methods" section) providing computed DMEs and PDAs relying on Eq. (2).

Figure 2 shows, for both experiment and theory, two-dimensional (2D) maps representing the magnitudes and phases of the PDAs for the parallel orientation resulting from the coherent superposition of the $(l, 0)$ partial-wave DMEs according

to Eq. (2), together with the 2D map of the photoionization time delays obtained as the energy derivatives of the PDA phases following Eq. (1), as a function of the photoelectron energy $E$ and the emission direction $\theta_k$. The reported PDA phases are obtained by referencing the $D_{lm}$ dipole matrix element phases to the $\eta_{2,1}$ phase as described above.

The common features of the measured and computed PDAs are highlighted by comparing the left and right halves of each of the 2D maps. Phases and time delays for an emission cone of 0–30° are attenuated in this visualization since the emission probability in direction of the N atom is quite low as visible in Fig. 2a and ref. [29] and phases are therefore less well-defined. Complementary one-dimensional plots featuring the energy dependence of the PDA magnitudes, phases, and time delays for a series of selected fixed emission angles between 45° and 180°, are reported in Fig. S4 (Supplementary Note 7), including uncertainties for the experimental results.

Experiment and theory demonstrate very similar features in the polar plot of the magnitudes of the PDAs, consistent with those observed in the MFPADs[29]. A distinct up–down asymmetry, characterized by a strongly favored emission probability towards the oxygen end of the NO molecule ($\theta_k = 180°$) at energies higher than 8 eV ($h\nu \geq 30$ eV), is observed within the structured pattern with four lobes reflecting the important contribution of the $f\sigma$ partial wave, which display a maximum in the 9–12 eV electron energy range.

The structure observed in the PDA magnitudes is also evident in the map showing the PDA phases in Fig. 2b and the photoionization time delays in Fig. 2c, demonstrating a significant MF anisotropy of the emission dynamics as well (see Fig. S4). The measured PDA phases, with specific mean values for the different angle sectors, display slope variations about the resonance, while the global phase change before or after the resonance ranges between 0 and $\pi$. For the lobes of maximum photoionization probability, around $\theta_k = 180°$, 120° or 60°, the measured time delay reaches +100 as on top of the resonance, while it is near zero or slightly negative in between. A remarkable structured energy profile is observed for the $\theta_k = 150–180°$ preferred emission cone, where the time delay passes through a minimum of −215 as at about 3–5 eV electron energy before rising up to +110 as at 9 eV, then flattening out to zero for higher energies. This overall behavior agrees qualitatively very well with the computed time delay energy profiles, although in the calculation the extreme values in this emission cone vary from −400 as to +160 as. A better quantitative agreement can be expected from calculations beyond the fixed nuclei approximation[23]. Large positive and negative group delays in the region of shape and autoionization resonances have also been observed in high-harmonic spectroscopy involving aligned $N_2$ molecules[36]. Localized discontinuities are observed, e.g., at ($\theta_k = 135°$; $E = 12–18$ eV) when the PDA goes through zero[35], together with a phase jump, similar to a Cooper minimum. Relying on the presented extraction of the complex-valued DMEs from measured MFPADs[37], in photoionization of small molecules, single-photon ionization delays, thereby reported with a complete angular and energy resolution for an orientation parallel to the polarization axis, are as well accessible for any orientation of the molecule and MF emission direction $\hat{k} \equiv (\theta_k, \phi_k)$.

**Behavior of MF angle-resolved PDA curves in a shape resonance.** The ionization delay energy profiles across the resonance for fixed MF emission directions $\theta_k$ can be rationalized by considering the energy dependence of the $D(\theta_k, E)$ PDAs as curves in the complex plane, resulting from the evolution of their real and imaginary parts with electron energy. This representation previously used to visualize the behavior of DMEs near Cooper minima[35] or the spectral phase profile of autoionizing resonances[10] is here exploited to feature the influence of the shape resonance on the energy dependence of MF angle-resolved channels. In Fig. 3 we present the PDA curves and the corresponding time delays for the experiment and calculations, at emission angles of 180° and 60°.

The evolution of the PDA in the complex plane with increasing energy is fairly similar in the experimental and computed data. The PDA at 180° describes nearly a full circle centered away from the origin in the third quadrant of the complex plane, yielding a very small overall phase change and therefore both large positive and negative time delays. For 60°, the PDA has shifted away from the origin in the first quadrant leading to an overall phase change smaller than $\pi$, and mostly positive time delays. A qualitative understanding of these different types of curves relying on a multichannel model of the behavior of the PDA across a resonance is discussed in the following section and further illustrated in Supplementary Note 8.

**Partial wave multichannel Fano line-shape analysis of a shape resonance.** The remarkable photoionization dynamics featured in Fig. 2 in the region of the $4\sigma \rightarrow k\sigma^*$ shape resonance can be interpreted based on a Fano line-shape analysis[38] as applied to multichannel problems[39–41]. The analysis models the contributions of resonant and non-resonant photoionization amplitudes that coherently add to yield the total PDA. The rich angular variation of the time delay is found to stem from the interference

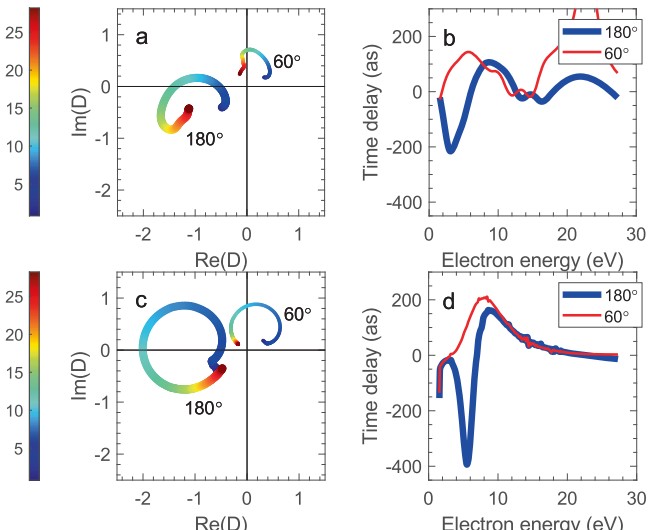

**Fig. 3 Experimental and computed PDA curves and time delays. a** PDA curves and **b** time delays for the experiment at emission angles of 180° (thick line, blue in **b**) and 60° (thin line, red in **b**). The corresponding data for the computed cross sections are given in (**c**) and (**d**). The color scale on the left gives the photoelectron energy in eV for the curves in the complex plane.

of the resonant and non-resonant paths, where the resonant path time delay is angle-independent.

In the original Fano analysis of photoionization with a resonance coupled to a continuum[38], the resulting line shapes were characterized by the position $E_{res}$, width $\Gamma$, and line-shape parameter, $q$. For a single resonance coupled to more than one asymptotic continuum, it was shown that the analysis could be transformed to that of the resonant state decaying to a single-channel plus a sum of non-interacting channels formed as linear combinations of the original channels. Later developments[39–41] provided expressions for the line shapes for decay to the original asymptotic channels, which when added together yield the same profiles as found in the original paper by Fano. Two-channel Fano-like parameterizations have recently been applied to predict the angular dependence of atomic photoionization time delays in the laboratory frame[14].

In the present study, implementing a multichannel analysis based on the expansion of the PDA as a coherent superposition of partial-wave channels (Eq. (2)), we show that the experimental and computed PDAs are very well fitted using a set of partial wave independent $E_{res}$ and $q$ parameters, and a total width $\Gamma$, defining a single resonant state.

For ionization from the $4\sigma$ orbital by light linearly polarized parallel to the molecular axis, with $m = 0$, $\mu = 0$, and $R_{0,0}^{(1)}(\hat{\Omega} = \hat{z}) = 1$, the PDA in Eq. (2) is written as

$$D(\theta_k, \varepsilon) = \sum_l D_l(\varepsilon) Y_{l0}(\hat{k}) = \sum_l D_l(\varepsilon) \sqrt{\frac{2l+1}{4\pi}} P_l(\cos \theta_k). \quad (3)$$

where $\varepsilon$ is the energy relative to the resonance energy $E_{res}$, scaled by the resonance width $\Gamma$

$$\varepsilon = \frac{E - E_{res}}{\Gamma/2}. \quad (4)$$

For such a state decaying into multiple partial waves, the $D_l(\varepsilon)$ DMEs can be expressed as the sum[41]

$$D_l(\varepsilon) = D_l^{(0)}(\varepsilon) + D_l^{res}(\varepsilon) \text{ for } l = 0, \ldots, l_x \quad (5)$$

where $D_l^{(0)}(\varepsilon)$ and $D_l^{res}(\varepsilon)$ are non-resonant and resonant contributions for each partial wave. The $D^{(0)}(\theta_k, \varepsilon)$ non-resonant and

**Table 1 Root-mean-square (RMS) deviation and fitting parameters of the Fano fit.**

|  | Experiment | Computation |
| --- | --- | --- |
| RMS fit | 0.014 | 0.011 |
| $E_{res}$ (eV) | 8.29 | 8.24 |
| $q$ | 2.709 | 1.690 |
| $\Gamma$ (eV) | 7.958 | 6.640 |
| $\Gamma_0$ (eV) | 0.014 | 0.043 |
| $\Gamma_1$ (eV) | 0.031 | 0.267 |
| $\Gamma_2$ (eV) | 1.121 | 0.114 |
| $\Gamma_3$ (eV) | 6.791 | 6.163 |
| $\Gamma_4$ (eV) | – | 0.053 |

Results obtained in the non-linear least-square fit of the experimental and computed PDAs with the multichannel Fano model (see text and the section "Methods").

$D^{res}(\theta_k, \varepsilon)$ resonant contributions to the $D(\theta_k, \varepsilon)$ PDA in Eq. (3) are then given by

$$D^{(0)}(\theta_k, \varepsilon) = \sum_l D_l^{(0)}(\varepsilon)\sqrt{\frac{2l+1}{4\pi}}P_l(\cos\theta_k) \qquad (6)$$

and (see the "Methods" section for details)

$$D^{res}(\theta_k, \varepsilon) = \frac{q-i}{\varepsilon+i}\sum_l \alpha_l(\varepsilon)\sqrt{\frac{2l+1}{4\pi}}P_l(\cos\theta_k). \qquad (7)$$

Performing a non-linear least-squares fit of the PDAs, either computed or obtained from the experiment, based on the above general form of $D^{(0)}(\theta_k, \varepsilon)$ and $D^{res}(\theta_k, \varepsilon)$, enables us to determine the resonance energy, $E_{res}$, the Fano line-shape parameter $q$, as well as the complex-valued $D_{l,n}^{(0)}$, i.e., the non-resonant DMEs, and the $V_l$ couplings between the resonant state and each partial wave directly related to the partial widths $\Gamma_l$ (see the "Methods" section).

The obtained fitting parameters are summarized in Table 1 and the subsequent separate resonant and non-resonant components of the transition are presented in Fig. 4 in terms of the magnitudes (a, b) and phases (c, d) of the PDAs, and the energy derivatives of the phases expressed as time delays (e, f), for both experiment (left half) and theory (right half). In the plot of the PDA magnitude for the resonant component (Fig. 4a), both show the dominant $f$-wave nature of the resonance, with large partial widths of the order of 6.5 eV for $l = 3$. For the non-resonant component of the transition, the measured and computed PDA magnitudes (Fig. 4b) display an asymmetry along the molecular axis favoring emission at $(\theta_k = 180°)$, reflecting the interference of $l\sigma$ partial waves of different parity. The phases (c, d), e.g. roughly out-of-phase and in-phase for the 0°–30° and 150°–180° emission cones, respectively, determine the coherent superposition of the resonant and non-resonant components of the PDA, with magnitudes described in (a, b), resulting in the PDA displayed in Fig. 2.

The nodal structures seen in the magnitudes in Fig. 4a and b lead to discontinuous jumps of $\pi$ in the phases depicted in Fig. 4c and d as the emission angle direction crosses such an angular node. When the emission angle location of a node has some energy dependence, i.e., in the radial direction as seen in Figs. 2b and 4d around 135°, then rapid phase changes by $\pi$ occur as a function of energy leading to the large positive and negative photoionization time delays in Figs. 2c and 4f.

The plot of the ionization delays for the resonant component (Fig. 4e) displays a positive maximum on top of the resonance of the order of 150 as for the experiment, and 250 as for the calculation, consistent with the trapping effect resulting from the

centrifugal barrier associated with the shape resonance. We stress that they display no angular dependence: this characteristic linked to the multichannel Fano model presented (see the "Methods" section) is validated by the excellent fit of the model to the reported results (Table 1). It is a reflection of the general observation in photoionization or electron scattering that for resonances narrower than changes in any non-resonant scattering in the vicinity of the resonance, one will measure a single lifetime, but possibly different amplitudes, for the different decay channels. We also note that in Fig. 4e the computed resonant time delay is always positive, whereas the experimental time delay, dominantly positive, reaches negative values for high energies. This feature can be assigned to the different energy dependence of the non-resonant $D_l^0(\varepsilon)$ included in the model fits. On the other hand, the corresponding non-resonant ionization delays display emission anisotropies assigned to the combination of the non-resonant scattering and the dipole coupling of the initial state to the continuum which can be analyzed in terms of the partial-wave composition of the $4\sigma$ orbital from which the photoelectron is ejected[42]. Some features common to the experimental and computed PDAs are visible, e.g., in the discontinuities corresponding to the PDA nodal structures around 135° in theory and 120° in the experiment (Fig. 4f), reflecting those observed in Fig. 2c. Consequently, within the multichannel Fano model, all the polar angle dependence in the time delay energy profiles reported in Fig. 2c for the full PDA originates from the interference between the resonant and non-resonant components of the PDA, where the ionization delay of the resonant term is angle-independent.

**Relationship between Wigner scattering time delay and photoionization time delay.** The analysis of the MF angle-resolved time delays in the presence of a shape resonance within the presented multichannel Fano model motivates a closer comparison of the resonant photoionization time delay, which displays no angular dependence in the MF, with a computed Wigner scattering time delay in the presence of resonance. We find that the resonant part of the photoionization time delay is comparable to half of the resonant Wigner scattering time delay.

First introduced in scattering theory, the so-called Eisenbud–Wigner–Smith time delay, or Wigner scattering time delay, defined as the delay of a wave packet scattering from a target relative to that of a free wave packet, is given by twice the energy derivative of the scattering phase shift[43,44].

For the resonant part of the PDA in the multichannel Fano fit, $D^{res}$, as defined in Eq. (7), the energy derivative of the phase leads to the photoionization time delay

$$\tau_{res} = \frac{2}{\Gamma}\frac{d\eta_{PDA}}{d\varepsilon} = \frac{2}{\Gamma}\frac{1}{1+\varepsilon^2} \qquad (8)$$

where the relative energy $\varepsilon$ is defined in Eq. (4) and we have assumed that there is no energy dependence in the phase of $\alpha(\varepsilon)$. Note that this is the same time delay as would be obtained from $D^{res}$ in the limit of the $D_l^0 \to 0$ (see the "Methods" section). This expression matches that corresponding to the energy derivative of the scattering phase shift, $\delta_{Scat}$, for a resonance fit by the Breit–Wigner form[45], or its multichannel generalization[32], which is

$$\frac{1}{2}\tau_{W-res} = \frac{2}{\Gamma}\frac{d\delta_{Scat}}{d\varepsilon} = \frac{2}{\Gamma}\frac{1}{1+\varepsilon^2} \qquad (9)$$

where $\tau_{W-res}$ is the resonant Wigner scattering time delay.

To illustrate this close correspondence between the resonant parts of the scattering Wigner time delay and the photoionization time delay for the studied reaction, we have computed the

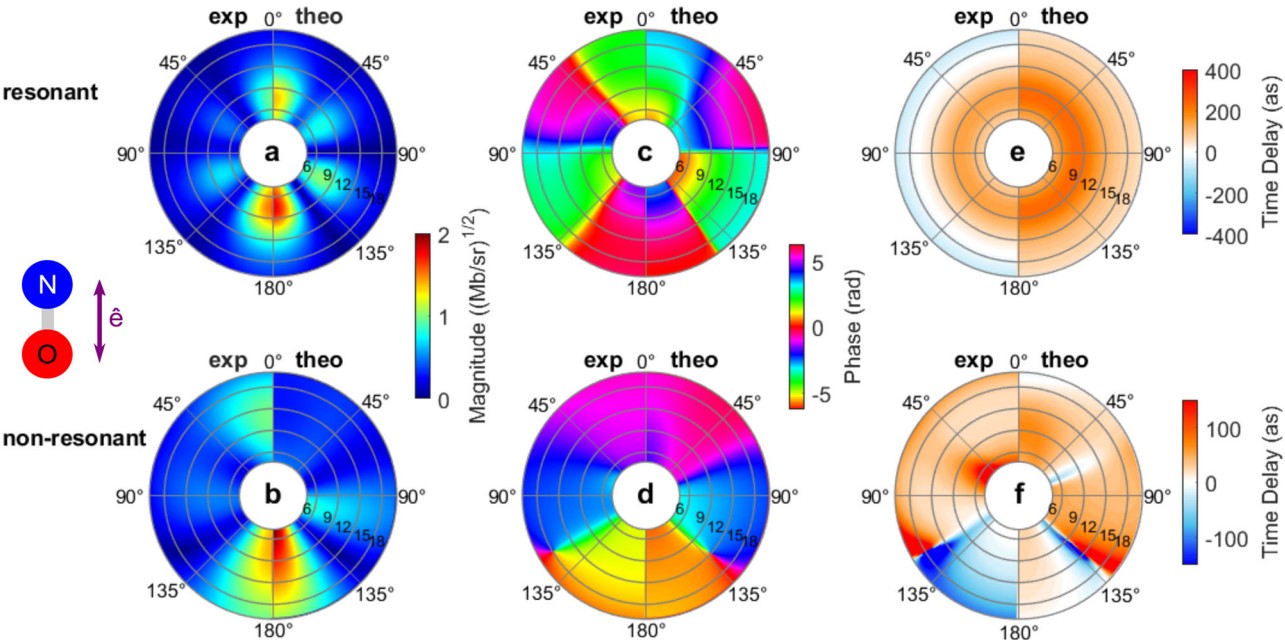

**Fig. 4 Resonant and non-resonant PDAs and photoionization delays.** Resonant (top) and non-resonant (bottom) PDA magnitudes (**a**, **b**), phases (**c**, **d**), and energy derivatives of the phases representing the time delays (**e**, **f**). The respective left (right) halves of the 2D maps show the data extracted from the experiment (theory). In the 2D polar plots, the radius indicates the photoelectron kinetic energy in eV, and the polar angle $\theta_k$ is relative to the molecular axis, with ($\theta_k = 0°$) pointing to the direction of the N atom. Note that the non-resonant time delays are on a reduced scale compared to the resonant ones. For both the computed and experimental data, we obtain a very good fit of the PDAs for $\theta_k$ between 0° and 180°, at electron kinetic energies from 4.65 eV to 17.05 eV: this energy range includes the shape resonance in the $c\,^3\Pi$ channel but excludes resonant contributions from other channels which strongly affect the PDAs below 4.65 eV. This supports the validity of the proposed multi-channel Fano model.

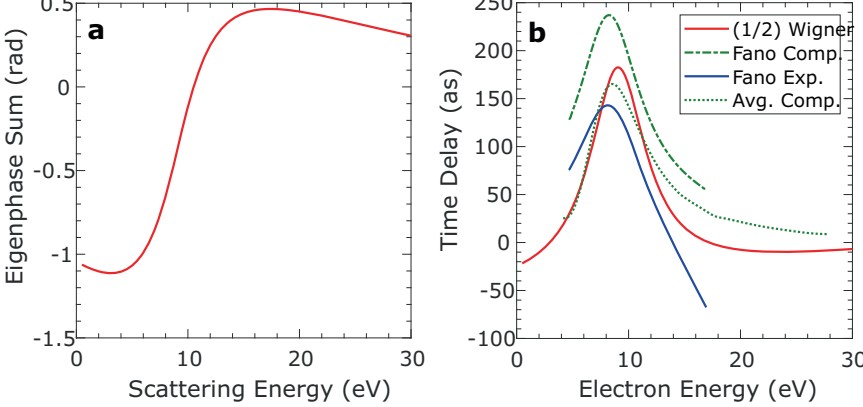

**Fig. 5 Eigenphase sum and Wigner time delay for scattering from NO$^+$.** Eigenphase sum for e-NO$^+$ scattering from a one-channel calculation including only the $c\,^3\Pi$ state of NO$^+$, with the electron-kinetic energies given relative to the threshold for that state (**a**). One channel $(1/2)\tau_{Wigner}$ time delay for the short-range part of the scattering phase shift compared to the photoionization time delays extracted from the resonant part of the multichannel Fano fit for theory and experiment as presented in Fig. 4e, and to the computed emission angle averaged ionization delay for the parallel orientation (see text) (**b**).

eigenphase sums[32] from the short-range scattering matrix for electron scattering from the NO$^+$ ion using similar computational methods as were used for the photoionization calculation (see Supplementary Note 5).

The fit of the computed eigenphase sum displayed in Fig. 5a using the Breit–Wigner form[32] leads to the resonance position, 9.09 eV, and width, 6.07 eV. When scattering also involves a non-resonant background component, as seen in the negative slope of the eigenphase sum before and after the resonance in Fig. 5a, the phase shifts associated with the background and with the resonance are simply added[32], so that their contributions to the Wigner scattering time delay are also additive. This results from the unitary scattering $S$-matrix and it differs from the situation

met in the evaluation of the photoionization time delay where the coherent sum of the resonant and non-resonant components of the PDA is considered. The half Wigner time delay reported in Fig. 5b is then a sum of the term coming from the pure Breit–Wigner form plus a non-resonant term. At the energy of 9.09 eV, the maximum value of $\frac{1}{2}\tau_W$, 182 as, results from the sum of 218 as (half of the resonant contribution) and −36 as (half of the non-resonant contribution).

Consistent with the results given in Eqs. (8) and (9), half of the resonant peak Wigner time delay is comparable to the computed photoionization time delay found in the resonant contribution to the PDAs, which has a peak value of 237 as displayed in Fig. 4e and reported in Fig. 5.

We note that the mean lifetime of the resonant state in the scattering resonance, using $\tau = 1/\Gamma$ (atomic units) with a width of 6.07 eV, gives a value of $\tau_W \approx 108$ as, close to the mean lifetime of the resonance extracted from the Fano model for the calculation, $\tau_{res}^T \approx 99$ as, and for the experimental data, $\tau_{res}^E \approx 83$ as. These values are also comparable to the computed emission averaged time delay for the considered parallel orientation, obtained as the average of the delay over emission directions, weighted by the differential cross sections[25].

This comparison shows that the resonant Wigner scattering time delay represents the main contribution to the resonant photoionization time delay at a shape resonance obtained in the multichannel Fano model. However, when considering the complete photoionization process, the non-resonant amplitudes, which combine the non-resonant scattering and the dipole coupling to the initial state, coherently add to the resonant amplitudes. This leads to the rich behavior in the angular dependence seen in the resulting PDA phases and corresponding photoionization delays.

## Discussion

In summary, we report experimental and computed photoionization delays for one-photon ionization of the NO molecule into the $NO^+(c^3\Pi)$ state across the $4\sigma \to k\sigma^*$ shape resonance. For the selected parallel transition experiment and theory demonstrate similar variations in the range of a few hundreds of attoseconds with electron energy and emission angle in the molecular frame.

Experimental photoionization delays, completely angle-resolved in the molecular frame for fixed-in-space targets, were obtained as the energy derivative of the PDA phases, accessible through MFPADs measured at a series of photon energies using synchrotron radiation, from which the partial wave complex-valued DMEs were extracted as described, and added coherently to form the PDAs. Ab initio calculations using the multichannel Schwinger configuration interaction method with 10 channels, performed at a fixed internuclear distance, compare well with those results up to the ionization delay level.

The analysis of the measured and computed complex-valued PDAs based on a multichannel Fano formalism enables us to separate the photoionization dipole amplitude into resonant and non-resonant contributions. Within this model, the observed MF angle dependence of the time delay profiles results from the coherent superposition of the resonant and the non-resonant components of the PDA, where the time delay of the resonant component is angle-independent. The in-depth description of the different contributions to the phase of the photoionization amplitude highlights the differences between the photoionization time delay and the corresponding Wigner scattering time delay.

In the future, these results could be fruitfully compared to angle-resolved two-photon ionization-time delays obtained in attosecond time-resolved experiments using, e.g., the RABBITT technique combined with coincidence electron–ion 3D momentum spectroscopy. One challenging difficulty to investigate a selected photoionization process, in that case, resides in the congested spectra resulting from photoionization by an harmonic comb into different ionic states, even in a small molecule[46]. Photoionization of NO into the $NO^+(c^3\Pi)$ state stands as a favorable prototype for this goal, as illustrated by the complete MFPADs produced by an XUV attosecond pulse train which were measured for the studied process[34].

The time delays reported here for one-photon ionization will also provide a benchmark to disentangle the contributions from the XUV and IR transitions to the measured two-photon

time-delays in a RABBITT scheme. For instance, stereo-Wigner time delays obtained in a RABBITT experiment could be compared to the here reported one-photon delays integrated over emission cones in opposite directions with respect to the orientation of the molecule.

Finally, relying on the achieved polar and azimuthal dependence of the PDAs and the related one-photon ionization delays when the cylindrical symmetry of the ionization process is broken, either for orientations of linear molecules other than parallel to the polarization such as the perpendicular orientation or for non-linear polyatomic molecules where photoemission delays are foreseen as a probe of molecular environment[47], will provide additional insights into the photoionization dynamics in both spectrally and time-resolved studies, including the role of resonances in these processes.

## Methods

**Experimental approach.** Experiments were performed at the XUV and VUV beamlines PLEIADES and DESIRS at the Synchrotron SOLEIL operated in the 8-bunch mode (50 ps pulses with a 147 ns period)[48,49] using circularly polarized light. MFPADs were measured using electron-ion coincidence 3D momentum spectroscopy based on a COLTRIMS-type set-up[50], taking advantage of dissociative photoionization (DPI) which results from the ejection of inner-valence shell electrons, leading to the production of an ionic fragment and a photoelectron. In this study of dissociative photoionization of NO, the velocity vectors $\mathbf{v}_{N^+}$ and $\mathbf{v}_e$ of N+ ions and photoelectrons are derived from the impact positions and times of flight (TOF) at the detectors, from which the MFPADs were constructed. The MFPADs are expanded using the $F_{LN}$-function formalism developed previously[33,51]. DMEs were then extracted by a non-linear least-squares fit of the experimental data at each energy as described in[33,51] giving access to the magnitudes $d_{l,m}$ and phases $\eta_{l,m}$ of the one-photon ionization dipole matrix elements. For further details on the experiments and data analysis, we refer to Supplementary Notes 1 and 2.

**Theoretical approach.** All dipole matrix elements for the photoionization of NO were computed in ten-channel calculations using the multichannel Schwinger configuration interaction (MCSCI) method described previously[29]. These calculations were based on a close-coupling expansion of the wave function of the photoionized system including a correlated representation of a selection of the lowest-lying ion states. For further details on the calculations, we refer to Supplementary Note 6.

**Multichannel Fano model for a resonance.** Setting $m = 0$, $\mu = 0$, and $R_{0,0}^{(1)}(\hat{\Omega} = \hat{z}) = 1$ in Eq.(2) for the studied parallel transition, the expansion of the PDA $D(\theta_k, \varepsilon)$ in partial-wave DMEs $D_l(\varepsilon)$ can be separated into non-resonant and resonant contributions, $D_l^{(0)}(\varepsilon)$ and $D_l^{res}(\varepsilon)$ (Eq. (5)) with

$$D_l^{res}(\varepsilon) = \alpha_l(\varepsilon)\frac{q - i}{\varepsilon + i}. \quad (10)$$

In this model, the Fano line-shape parameter $q$ is the same for all emission directions, and the quantity $\alpha_l(\varepsilon)$ is related to the strength of the $V_l$ coupling between the resonant state and each scattering partial wave $l$, which is assumed to be energy-independent, as follows:

$$\alpha_l(\varepsilon) = V_l\left[\frac{2\pi}{\Gamma}\sum_{l'=0,\ldots,l_x}D_{l'}^{(0)}(\varepsilon)V_{l'}^*\right], \quad (11)$$

The total width is the sum of the partial widths $\Gamma_l$

$$\Gamma = \sum_l \Gamma_l, \quad (12)$$

where $\Gamma_l$, proportional to the rate of decay of the resonance into the $l$ partial-wave channel, is related to the $V_l$ coupling by

$$\Gamma_l = 2\pi|V_l|^2, \quad (13)$$

Note that the energy derivative of the phases of the $D^{res}(\theta_k, \varepsilon)$ and thus the resonant time delays, as presented here, do not have any angular dependence. This is a result of the assumption that the coupling matrix elements $V_l$ are energy independent [see Eqs. (7) and (11)].

We also note that the square of the transition matrix element if all the non-resonant $D_l^0$ go to zero, with $\alpha_l(\varepsilon)$ also approaching zero and increasing values of q, would result in a Lorentzian function with a width of $\Gamma/2$.

In applying these formulas to the interpretation of the MFPADs in NO photoionization the non-resonant DME magnitudes $D_l^{(0)}(\epsilon)$ are assumed to be

slowly varying with energy and described by a quadratic form

$$D_l^{(0)}(\varepsilon) = \sum_{n=0,1,2} \varepsilon^n D_{l,n}^{(0)}. \tag{14}$$

## Data availability

All datasets generated in this study have been deposited in the open-access repository Zenodo: https://doi.org/10.5281/zenodo.5585867.

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

## Acknowledgements

We gratefully acknowledge J. Bozek PLEIADES-SOLEIL beamline manager, C. Nicolas and A. Milosavljevic, PLEIADES beamline scientists, E. Robert, beamline engineer assistant on PLEIADES beamline, and L. Nahon, DESIRS-SOLEIL beamline manager, G. Garcia, DESIRS beamline scientist, J.F. Gil, beamline engineer assistant on DESIRS beamline, for their cooperation and support. We thank the SOLEIL general staff for smoothly operating the facility. We are grateful to K. Veyrinas, M. Hervé, and M. Gisselbrecht for their contribution to the experiment. We thank J. Guigand, S. Lupone, N. Tournier, O. Moustier (ISMO) for technical support in maintenance of the CIEL set-up. R.R.L. and D.D. acknowledge fruitful discussions with P. Salières. D.D. is very grateful for stimulating discussions with T. Jahnke within the ASPIRE ITN. This work is supported by "Investissements d'Avenir" LabEx PALM (ANR-10-LABX-0039-PALM), EquipEx ATTOLAB (ANR-11-EQPX-0005-ATTOLAB), and ASPIRE Marie Sklodowska-Curie ITN (EU-H2020 ID: 674960). The theoretical research performed at Lawrence Berkeley National Laboratory was supported by the US Department of Energy, Office of Science, Office of Basic Energy Sciences, Chemical Sciences, Geosciences, and Biosciences Division, under Contract No. DE-AC02-05CH11231. This research used the resources of the National Energy Research Scientific Computing Center, a US Department of Energy Office of Science User Facility, and the Lawrencium computational cluster resource provided by the IT Division at the Lawrence Berkeley National Laboratory.

## Author contributions

F.H., J.J., M.L., J.C.H., and D.D. performed the measurements at the synchrotron beamlines. F.H., J.J., and J.C.H. analyzed the experimental data. R.R.L. and F.H. developed and used the fitting routines. R.R.L. developed the theoretical models and

calculations. D.D. supervised the project. The manuscript was written by F.H., D.D., and R.R.L.

## Competing interests
The authors declare no competing interests.

## Additional information

**Peer Review Information** *Nature Communications* thanks the anonymous reviewers for their contribution to the peer review of this work. Peer reviewer reports are available.

