## [Peer Review File · Nature Communications]

Influence of Shape Resonances on the Angular Dependence of Molecular Photoionization DelaysReviewers' comments:

Reviewer #1 (Remarks to the Author):

The paper by F. Holzmeier et al presents interesting and well-described new theoretical and experimental results concerning the interference between resonant and non-resonant contributions in the region of a continuum resonance (the shape resonance) in NO.

The experimental and theoretical data are undoubtedly well presented and convincing.

However, the “photoionization time delay” framework is quite an overstretch in my opinion. While there exists a relevant bulk of literature concerning photoionization time delays, mainly by RABBITT and related techniques able to reach the attosecond timescale, in the present work photoionization time delay is not obtained by time-resolved tools, but it is a matter of definition: in fact, it is defined as the energy derivative of the phase of the angle-dependent photoionization dipole amplitude.

This is a legitimate choice, but then photoionization time delay sounds more like a “catching” label than a real time-resolved result. I myself judging on the title was naively expecting a paper with measurements performed at an HHG source, not at a synchrotron. In other words, it can be misleading for readers looking for results obtained on a true attosecond timescale, especially because a paper published on Nature Communication is supposed to address a wider audience than the scientific community closer to the work illustrated.

A complementary criticism to the methodology presented here is also reported in the paper by P. Hockett et al, quoted as Ref. 25 in the present work. The authors there specify that the method of determining the scattering phases of individual partial waves by measurements of photoelectron angular distributions is rather limited, since it is based on a reference wave and cannot give a full mapping of the time delay.

I am not questioning the validity of the results here described, which definitely deserve publication elsewhere. However, in my opinion this paper does not match the criteria of general interest of Nature Communications. I suggest to transfer it to Scientific Reports, or to resubmit it to a more specialized journal. I further suggest to possibly change the title and “tone down” the claim on photoionization time delays, or at least to be more explicit about the limitations of the present approach.

Reviewer #2 (Remarks to the Author):

This manuscript presents a combination of extensively analyzed experimental work and theoretical modeling focusing on photoionization time delays, in particular on their angular dependence. The abstract claims that this work "...completely resolves experimental and computational angular dependence of single-photon ionization time delays in NO molecules in a shape resonance". I am not exactly sure what that means, as the agreement between the presented experimental and theoretical results are rather poor and my confidence in validity of both is low. It goes on to state its apparent main result in the following sentence: "The remarkable time delay variations are found to originate from the interfering angle-independent resonant and angle-dependent non-resonant contributions to the dynamics of the ejected electron". Again, I am not sure why those time delay variations are characterized as "remarkable", nor am I impressed by the attribution of those variations to "interference" of various "contributions to the dynamics of the ejected electrons". That statement is rather obvious and therefore not very meaningful – of course, any quantity related to photoionization is almost by definition originates from interference of different contributions. That is not the type and level of insight or conclusions which would get a work published in Nature Communications. And that is just in the abstract. It does not get any better from there. The manuscript is rather haphazardly put together – to get through the presentation and grasp its main ideas a reader is referred to no less than 10 Supplementary notes. Obviously, otherwise the authors would not have met the size requirements. This makes the reading and understanding this work quite a chore. And after I went through the main text and the supplementary notes a few times I was still not quite sure why this manuscript was even written and why its authors think that it deserves acceptance in Nature Communications. In the following I will try to explain why I do not share the authors' enthusiasm for this work and why my advice is to reject it.

This manuscript combines two parts – "experimental" and theoretical. The "experimental" part starts with an excellent experiment on measurements of angular resolved photoelectron spectra of NO using synchrotron radiation at 11 different photon energies with a step of 1.55 eV across the shape resonance at 31 eV. With ionization leading to dissociation and with this being a momentum-resolved electron-ion coincidence measurement, the resulting massive array of data is extremely rich – it consists of full angle-resolved electron momentum spectra for all possible orientations of the molecules in Laboratory Frame. With the next step (described in Supplementary Note 3) the authors extract molecular frame photoelectron angular distributions (MFPADs). Should they left it at that, compared the measured MFPADs with theoretically calculated ones and discussed level of agreement between the two and possible reasons for any discrepancies that would have been a very fine and useful paper – though still not of the Nature Communications level. But in their attempts to raise the level of significance of their work by making it about "photoemission delays" the authors pile on four more levels of analysis – (1) they extract dipole matrix elements (DME) from the MFPADs; (2) then extract photoionization dipole amplitudes (PDA) from DMEs; (3) then extract the phase of those complex-valued PDAs η and (4) finally extract photoionization time delays τ by taking a derivative of η in respect to photon energy. That is just how one finds a group delay for a wavepacket – by taking a derivative of its phase in respect to energy –

and it works well if the wavepacket is not too far from a Gaussian one. But with the experimental energy spacing being rather sparse, each step of this rather convoluted extraction procedure relies on complicated fitting procedures and other approximations which are prone to introduce significant errors and uncertainties. The combined validity of such “extraction” procedure is rather dubious, in my opinion. Even that would have been fine, perhaps, if there were some attempts at independent validation of the extracted values for the photoemission delays, i.e. by comparing them with values obtained by a more direct time-resolved measurement such as streaking. Absent such attempts there is no guarantee that the “extracted” values are not completely meaningless. Obviously, by feeding some measured numbers into a convoluted multi-step “extraction” algorithm, some “extracted” numbers will come out of the other end. But why should we trust them? OK, maybe we could trust them if the authors manage to reproduce them with purely ab initio theoretical modelling performed at sufficiently high level which we have no reason to doubt. But even in that case I see no reason to go beyond MFPADs comparison. If measured and calculated MFPADs agree, so should the delays. But what is the point of calculating the delays, beside making the work look sexier? Neither experiment nor theory are time-dependent – they both rely on taking the numerical derivative to produce the delays. Moreover, the agreement between the experiment and theory is not so impressive either – it is qualitative at best. What do we learn from all that? Not much, in my opinion, and certainly not enough to support publication of this manuscript in Nature Communications.

My advice to the authors would be to forget about the time delays, to rewrite the manuscript focusing on MFPADs comparison and to submit it to some other journal – PRA seems appropriate.

Reviewer #3 (Remarks to the Author):

The submission "What Causes the Angular Dependence of Photoionization Time Delays in a Shape Resonance?" by Holzmeier et al. describes a new experimental approach to measure photoemission delays, utilising a reaction microscope to measure full molecular-frame photoelectron angular distributions at various photoionisation energies. In particular the authors scanned across a shape resonance in the NO molecule. Using a Fano line-shape analysis the authors could disentangle resonant and non-resonant contributions to the photoionisation dipole amplitude and hence ionisation delay.

The manuscript describes a new experimental approach to measure photoemission delays for molecular systems, which to my knowledge has not been previously demonstrated. The topic of electron dynamics during photoemission is of considerable interest across the wider physics community at the moment. The presented new methodology has several advantages over established methods, one of the key ones being that it does not rely on streaking techniques and attosecond light sources. This means that one can avoid the inherent large bandwidth of attosecond pulses and hence be much more selective with

respect to the ionisation/excitation process. Using narrow-bandwidth sources is particularly important as one starts to investigate more complex molecular systems, with an inherently higher density of states. Furthermore, the employed multichannel Fano analysis allows the authors to nicely separate the resonant and non-resonant dynamics, clearly showing which partial waves dominate the resonance behaviour and recovering the isotropic nature of the emission delay on resonance. I believe this is a novel and very powerful approach beyond the current state-of-the-art, that overcomes several challenges with existing techniques. The ability to address specific molecular resonances, and disentangle the resonant and non-resonant behaviour, is particularly exciting.

I do have a few comments I would like the authors to address:

-In Fig4b, showing the non-resonant PDA magnitudes, the theoretical modelling does not seem to show any intensity in the backwards (0 degree) direction. Further, the secondary maximum is not at 90 degrees (as in the experimental data) but shifted somewhat towards larger angles - which is somewhat unintuitive for me. Could the authors comment on this?

-In the second to last paragraph on page 7 (l37-47) the figure labels are wrong and should presumably refer to figure 5. Further, the following statement: "When scattering also involves a non-resonant background component, as seen from Fig.8(a), the phase shifts associated with the background and with the resonance are simply added, so that their contributions to the Wigner scattering time delay are additive." - to me it is not clear how the figure shows the additive nature of the phase shift.

-Generally the article is quite long and at times perhaps a bit too technical for the general audience of Nature Communications. The authors could consider placing certain parts within the supplementary information, e.g. "Behaviour of PDA curves in a Resonance", or parts of the derivation of the Fano Line-Shape analysis.

Small notes/typos:

-PDA on first use undefined (p2, l16)

-Fig. 3b/d the colour scale of the line is unnecessary and makes the two traces harder to distinguish. Perhaps just use different coloured lines here.

-Reference to Eq. 24/25 should presumably refer to references 24/25 (p7, l48)

Once these comments have been addressed I believe the paper is well-suited for publication in Nature Communications.

Referee 1:

We gratefully acknowledge the referee for his/her general evaluation of the work reported.

“However, the “photoionization time delay” framework is quite an overstretch in my opinion. While there exists a relevant bulk of literature concerning photoionization time delays, mainly by RABBITT and related techniques able to reach the attosecond timescale, in the present work photoionization time delay is not obtained by time-resolved tools, but it is a matter of definition: in fact, it is defined as the energy derivative of the phase of the angle-dependent photoionization dipole amplitude.

This is a legitimate choice, but then photoionization time delay sounds more like a “catching” label than a real time-resolved result. I myself judging on the title was naively expecting a paper with measurements performed at an HHG source, not at a synchrotron. In other words, it can be misleading for readers looking for results obtained on a true attosecond timescale, especially because a paper published on Nature Communication is supposed to address a wider audience than the scientific community closer to the work illustrated.”

There are two aspects raised by the referee in this first comment. (i) Is the definition used to qualify the measured observables as photoionization delays a “catching label”? Is it an overstretch to speak about “one-photon ionization delays” for entities obtained in an experiment using synchrotron radiation, although accessing “photoionization delays” has become a major goal in RABBITT and related time-resolved techniques reaching the attosecond time scale, as attested by a number of relevant publications (see e.g., ref. [7-23] cited in the introduction of our manuscript)? (ii) Is this misleading for readers of Nature Communications interested in the photoionization dynamics at such time scales?

(i) As the referee writes, the definition of “time delay” that we use as the “energy derivative of the phases of photoionization amplitudes” is legitimate, and it is even the historical definition of this concept given by Eisenbud-Wigner-Smith (EWS) in their seminal work addressing scattering time delays around 1955-1960, which were defined as the energy derivative of the scattering phase shifts (and most often referred to in recent literature). A related time delay, but in general different from a strict scattering delay, that can be measured in a photoionization experiment, is the time delay of a wave packet formed by a short light pulse compared to some reference wave packet. Using first-order perturbation theory, the photoionization time delay is as well determined by the energy dependence of the phases of the photoionization transition amplitudes, again relative to some reference set of amplitudes. The difference between scattering and photoionization delays is discussed in the last section of the manuscript.

One-photon photoionization delays characterizing photoionization dynamics in molecules at the most fundamental and complete level (i.e., in the molecular frame, MF) are indeed obtained in the present work, therefore we do not believe that it should be felt as an overstretch.

(ii) One could anticipate that readers of Nature Communication, who might expect measurements performed at an HHG source since many recent publications address this goal in time resolved studies using ultrashort pulses, would not feel misled, but rather positively surprised and therefore interested, to realize that the photoionization dynamics can indeed be quantified at the attosecond level in the molecular frame using spectrally resolved experiments at the synchrotron radiation, and a methodology whose validity is discussed in the manuscript. In order to make our experimental

approach clear from the beginning we now mention in the abstract that synchrotron radiation and time-independent ab initio methods are exploited to determine photoionization delays.

For completeness, we may also add that, although in attosecond time-resolved experiments, such as RABBITT or streaking, the data points are recorded in scans with sub-femtosecond steps, time delays are as well obtained as energy derivatives of the phase shifts of the different side bands or streaking traces, respectively.

Furthermore, it was recognized recently (page 1/line 47 and ref. [20,25]) that, as a consequence of the non-spherical character of the molecular potential, attosecond time-resolved methods such as streaking and RABBITT, where ionization is induced by an XUV pulse and probed by a superimposed IR field, measuring thus *two-photon* ionization delays, *do not give access* to the *one-photon* XUV delays. Note that for atoms, by contrast, this can be achieved in specific conditions and approximations. Additionally, the broadband character of the ultrashort XUV pulses generates congested photoelectron spectra, due to the number of molecular ionic states and their intrinsic width when repulsive potential curves are involved, which in most cases prevents the extraction of dynamical features for ionization into a given ionic state, in particular for inner-valence ionization of molecules.

In contrast, in our study we take advantage of spectrally resolved measurements using synchrotron radiation to obtain the complex-valued photoionization dipole amplitude and can therefore access one-photon MF resolved photoionization delays.

“A complementary criticism to the methodology presented here is also reported in the paper by P. Hockett et al, quoted as Ref. 25 in the present work. The authors there specify that the method of determining the scattering phases of individual partial waves by measurements of photoelectron angular distributions is rather limited, since it is based on a reference wave and cannot give a full mapping of the time delay.”

Indeed, as noted by referee 1, using the method described in our manuscript, the determination of photoionization delays relies on the choice of a reference phase which establishes the coherence between the energies scanned across the shape resonance. Here we chose the reference phase from one of the non-resonant partial wave dipole matrix elements, and this step is a priori clearly stated in our manuscript, while providing scientific arguments and criteria which legitimize the choice made (see page 3/line 13 and SM Notes 4 & 5).

The referee refers to the paper by P. Hockett et al. (ref. [26] in our revised manuscript), in which it is indeed stated that the type of measurements that we performed is “typically not able to ascertain the phase structure with respect to energy, so can only determine η_{lm} for a given set of partial-waves with one of the waves serving as a reference. These types of measurement therefore provide detailed information on the angular part of the problem, including the phases of the contributing partial-waves, but do not directly provide a full mapping of $\tau_w(k, \theta, \phi)$.” Interestingly, in the ArXiv version of this manuscript ([1512.03788.pdf \(arxiv.org\)](https://arxiv.org/pdf/1512.03788.pdf)), including the latest revision from February 2017, i.e., after the publication of the actual J. Phys. B article, this paragraph is and has always been complemented by a footnote: “Although this is strictly correct, it is the case that careful analysis of PADs recorded at different energies, can provide phase information as a function of energy. [...] With such an approach the full energy and angle-dependent $\tau_w(k, \theta, \phi)$ could be obtained.” By selecting a partial wave, which does not contribute to the parallel transition and the shape reference, as our reference, we obtain such a full energy and angle-dependent time delay including all of the effects of the resonance in the parallel photoionization transition.

For completeness, we also emphasize that when performing time resolved experiments, the measured two-photon emission delays also involve an explicit reference: time delays were reported, e.g., for ionization of a given atomic orbital relative to another one (e.g., 3s and 3p in ionization of Ar in the pioneering experiments by Klünder et al., ref. [8]), or delay for emission at a given angle in the lab frame relative to a reference emission direction (Heuser et al., ref. [12]), difference in emission time delays on both ends of a molecule (Vos et al., ref. [27]).

These considerations do not lead us a priori to remove the reference to photoionization time delays in the title of the manuscript, since we are dealing with the proper definition of this concept, and we address indeed their determination from the energy derivative of phases, as do time-resolved techniques with attosecond pulses up to now. These results support the second part of the manuscript: addressing the origin of the observed angular dependence by the Fano multichannel approach describing the interference of resonant and non-resonant channels, tagged by a distinct angular dependence.

Referee 2:

The report of referee 2 is more difficult to answer in a rigorous and constructive way since it expresses a series of comments and judgments based openly on a lack of confidence in the validity of the reported experimental and theoretical results, a misinterpretation and/or belittlement of some statements, as for example the way the content of some sentences of the abstract are emptied of their meaning by omitting key words which describe precisely the scientific content of the results presented.

Let us stress that the scientific goal addressed in our manuscript is to report and interpret the observed strong variation of one-photon ionization delays amounting to few hundreds of attoseconds as a function of both the emission direction in the molecular frame and the photoelectron energy across a shape resonance (that we characterized as “remarkable” in the abstract).

As an example, the sentence quoted by referee 2: “The remarkable time delay variations are found to originate from the interfering **angle-independent resonant** and **angle-dependent non-resonant** contributions to the dynamics of the ejected electron”, is reduced to “interference of various contributions to the dynamics of the ejected electron” in his/her report. This shortened version is naturally “rather obvious” because most quantum effects are due to some type of interference. Here we identify *which* interference effects cause the angle dependent time delays. This important part of the work underlining the origin of the MF angular dependence of the time delays, through an original application of a Fano multichannel analysis, is absent in the report.

We aimed at organizing the manuscript in a comprehensible structure and we made use of supplementary notes to facilitate access of the readers to relevant methodological tools avoiding an extensive use of the literature and to provide illustrative complementary presentation of the results (e.g, 1D versus 2D). The notes are meant to provide the interested reader more insights into the methods (Notes 1-4,6), a visualization using the theoretical data of the effect of using relative phases (Note 5), a complementary 1D visualization of the results (Note 7), and a simple model that helps the non-expert reader to retrace our interpretation (Note 8). We do not think that the amount of supplementary information we provide is inappropriate compared to other articles published in Nature Communications or journals with a similar scope and readership. However, to account for this assessment, we have suppressed two notes from the Supplementary information.

Additional points raised by referee 2 are now considered:

This manuscript combines two parts – “experimental” and theoretical. The “experimental” part starts with an excellent experiment on measurements of angular resolved photoelectron spectra of NO using synchrotron radiation at 11 different photon energies with a step of 1.55 eV across the shape resonance at 31 eV. With ionization leading to dissociation and with this being a momentum-resolved electron-ion coincidence measurement, the resulting massive array of data is extremely rich – it consists of full angle-resolved electron momentum spectra for all possible orientations of the molecules in Laboratory Frame. With the next step (described in Supplementary Note 3) the authors extract molecular frame photoelectron angular distributions (MFPADs). Should they left it at that, compared the measured MFPADs with theoretically calculated ones and discussed level of agreement between the two and possible reasons for any discrepancies that would have been a very fine and useful paper – though still not of the Nature Communications level. [...]

My advice to the authors would be to forget about the time delays, to rewrite the manuscript focusing on MFPADs comparison and to submit it to some other journal – PRA seems appropriate.

We acknowledge the referee for his/her suggestion: however, as is cited in the manuscript, we have published earlier experimental and theoretical MFPADs studies of this process, first at a fixed photon energy (Lebech et al., ref. [33]), and more recently at selected photon energies illustrating the influence of the shape resonance on MFPADs (Veyrinas et al., ref. [29]) (and parallel studies for others processes and molecules). The interesting step addressed here, which we consider as very timely and of broad interest as well to the attosecond community, is to exploit the photoionization delays that are encapsulated in complete spectrally resolved MFPAD measurements, and range in the attosecond time scale. Here we proceed through the determination of the partial wave dipole matrix elements (DMEs), relying on one methodological assumption which is described and assessed in the manuscript. The angular and energy dependence of the ionization time delays is the observable which enables us to rationalize the effective role of the shape resonance based on a Fano multichannel approach: the shape resonance is found to induce positive time delays featuring the trapping of the photoelectron, and to strongly modulate the electron dynamics in terms of emission angle and energy, but it is not the origin of the angular dependence of the time delays as such.

But in their attempts to raise the level of significance of their work by making it about “photoemission delays” the authors pile on four more levels of analysis – (1) they extract dipole matrix elements (DME) from the MFPADs; (2) then extract photoionization dipole amplitudes (PDA) from DMEs; (3) then extract the phase of those complex-valued PDAs η and (4) finally extract photoionization time delays τ by taking a derivative of η in respect to photon energy. That is just how one finds a group delay for a wavepacket – by taking a derivative of its phase in respect to energy – and it works well if the wavepacket is not too far from a Gaussian one. But with the experimental energy spacing being rather sparse, each step of this rather convoluted extraction procedure relies on complicated fitting procedures and other approximations which are prone to introduce significant errors and uncertainties. The combined validity of such “extraction” procedure is rather dubious, in my opinion. (...) Obviously, by feeding some measured numbers into a convoluted multi-step “extraction” algorithm, some “extracted” numbers will come out of the other end. But why should we trust them?

Because the MFPADs result from the interference of a number of partial waves, it is possible through a thorough non-linear fit of the MFPADs (as described) which represent emission probabilities, to capture the magnitudes of a series of n DMEs and their $(n-1)$ relative phases. The extraction of complex valued DMEs from MFPAD measurements was discussed around 2000 by Cherepkov and coworkers as quoted in the manuscript (ref. [35], as well as in our earlier work (ref. [33])). There, as recalled in the Supplementary notes 3 and 4, the complete expansion of the MFPAD measured with circularly polarized light in CL’LN coefficients, combined with a non-linear fit of an extended set of MFPADs derived from these coefficients expressed in terms of DMEs, provides a solid ground for the extraction of the partial wave dipole matrix elements which constitute the building blocks of the photoionization amplitude. This describes the level (1) in the report of referee 2, where the degree of confidence in the evaluation of the DMEs is mostly described by the statistical uncertainties reported on the one-dimensional plots in Figure S1, deriving from the standard deviations provided by the Levenberg-Marquardt algorithm. The three other “levels”, presented in the report as piling up “complicated fitting procedures and other approximations prone to introduce significant errors and uncertainties” are streamlined steps in the exploitation of the DMEs to obtain the photoionization amplitude, its argument and energy derivative, leading to the time delays, as described by Equations (2) and (1), and related propagation of uncertainties.

In “level (2)” a spline fit connecting the 11 points, plus 2 additional energies, are justified by the continuous energy dependence deriving from the broad character of the studied shape resonance and supported by the multi-channel Schwinger configuration interaction theoretical method, which stands as one of the most accurate theory to account for molecular photoionization (often qualified as “cutting-edge calculations” in the literature). We agree that measuring MFPADs with a smaller energy spacing (equal to 1.5 eV in this work), which is at hand using synchrotron radiation, would add value to the experimental results, but the uncertainty analysis performed and the theoretical results support the conclusion that it would not change the major conclusion obtained for the time delays, their magnitudes and their angular dependence. At this point, we also emphasize that in a standard RABBITT experiment using HHG from a fundamental laser at 800 nm, the energy spacing between two successive harmonics in the attosecond pulse train, involved in the two pathways which interfere to produce the sideband signal, is 3.1 eV, i.e., twice the energy spacing considered in our experiment.

Concerning the iterative question of “trust” raised by referee 2, we deem that this question in an experimental study relies on the description of the data analysis and the evaluation of uncertainties, what belongs to the work presented. Nevertheless, in the proposed revised version, we emphasize further the difference between the DME extraction level (1), and the forthcoming steps to avoid any ambiguity, and we added a few comments on the propagation of uncertainties.

Even that would have been fine, perhaps, if there were some attempts at independent validation of the extracted values for the photoemission delays, i.e. by comparing them with values obtained by a more direct time-resolved measurement such as streaking. Absent such attempts there is no guarantee that the “extracted” values are not completely meaningless. Obviously, by feeding some measured numbers into a convoluted multi-step “extraction” algorithm, some “extracted” numbers will come out of the other end. But why should we trust them?

This statement is puzzling to us. If such a comparison had been within reach, we would obviously have discussed it in our work! However, this is not possible for two reasons. (i) To our best knowledge, there exists at this moment no access to a RABBITT or a streaking experiment reporting time resolved MFPADs, thereby accessing MF angle resolved photoionization delays, except for the determination of stereo time delays through a RABBITT study of inner valence ionization of CO, as quoted in the introduction of the manuscript [ref. 27], which compares emission characteristics within two large angular cones. (ii) Even if such measurements would be performed, only two-photon ionization delays could be determined, as stated in the introduction of the manuscript. Therefore, a comparison with the present results would certainly be of interest, but the interpretation would not be as simple as seems to be implied by referee 2. It would thus not constitute a “guarantee”, rather an interesting source of complementary knowledge that we foresee to investigate.

OK, maybe we could trust them if the authors manage to reproduce them with purely ab initio theoretical modelling performed at sufficiently high level which we have no reason to doubt. But even in that case I see no reason to go beyond MFPADs comparison. If measured and calculated MFPADs agree, so should the delays. But what is the point of calculating the delays, beside making the work look sexier? Neither experiment nor theory are time-dependent – they both rely on taking the numerical derivative to produce the delays. Moreover, the agreement between the experiment and theory is not so impressive either – it is qualitative at best.

This comment is again very difficult to rationalize and understand: one really wonders what is the scientific basis of the systematic doubt expressed by referee 2, here concerning the calculation? The reported MCSCI results (described in SI note 6) are state-of-the-art *ab initio* calculations, broadly acknowledged in the community interested in photoionization of molecules. Within the approximations used, e.g., fixed nuclei, the specified one-electron basis set, the specified close-coupling calculation with correlated target states, we have very high confidence that results are correct. We already reported an overall good agreement between the measured and computed MFPADs for the studied process at several photon energies (ref. [29,33]). Here the comparison between experiment and theory is extended to consider the complex valued DMEs for seven partial waves, and subsequent photoionization dipole amplitudes (PDAs) for all emission directions. These quantities are then compared over a broad energy range, across the shape resonance. To facilitate this comparison, one-dimensional magnitudes and phases of the PDAs, as well as the derived photoionization delays are reported in Fig. S2, together with related uncertainties for a series of selected emission angles. They illustrate a reasonably good prediction of the observables. As we mention in the manuscript, one interesting extension of these calculations would be to consider a range of internuclear distances about the equilibrium geometry of the NO(X, v=0) ground state and see the effects at the level of time delays at the maximum of the resonance.

The second theoretical component of the work is the development of a Fano multichannel model, which aims at separating the contributions of the resonant and non-resonant components of the studied photoionization up to the dynamics quantified by the time delays. This aspect of the work, which leads to the main conclusion concerning the origin of the angular dependence of the ionization delays, precisely contributes to answer the final question addressed in the report “What do we learn from all that?”

Referee 3:

We would like to thank Referee 3 for his very constructive feedback. His assessment is very positive and specific points are raised which we have addressed to improve the manuscript.

In Fig4b, showing the non-resonant PDA magnitudes, the theoretical modelling does not seem to show any intensity in the backwards (0 degree) direction. Further, the secondary maximum is not at 90 degrees (as in the experimental data) but shifted somewhat towards larger angles - which is somewhat unintuitive for me. Could the authors comment on this?

This question is quite difficult to answer. The agreement between experiment and theory in the 0° direction for the non-resonant magnitudes shown in Fig. 4b is indeed worse than for other emission angles. This figure shows the result of our multichannel Fano-model fit, which reproduces the overall experimental and computed PDA very well as seen from the RMS of the fit listed in Table 1. Since the total emission in this direction is low, it might be possible that this effect could originate from higher l partial waves that have a low overall weight and become only significant in this emission direction, where the dominant partial waves destructively interfere. The angle 90° has no specific meaning for the PDA since the molecule does not have a center of symmetry, so in our opinion a behavior of the non-resonant part of the PDA cannot be predicted. The conclusions presented in our manuscript rely mostly on the phases and time delays. We therefore choose not to discuss in detail disagreements between experiment and theory for the magnitudes for emission directions with very low flux.

In the second to last paragraph on page 7 (137-47) the figure labels are wrong and should presumably refer to figure 5. Further, the following statement: "When scattering also involves a non-resonant background component, as seen from Fig.8(a), the phase shifts associated with the background and with the resonance are simply added, so that their contributions to the Wigner scattering time delay are additive." - to me it is not clear how the figure shows the additive nature of the phase shift.

The referee is right that we had several errors with our references in this paragraph:

Page 7, line 37 was corrected to: "The fit of the computed eigenphase sum displayed in Fig. 5a using..." The following sentence was also rewritten in a clearer way and now reads: "When scattering also involves a non-resonant background component, as seen in the negative slope of the eigenphase sum before and after the resonance in Fig. 5a, the phase shifts associated with the background and with the resonance are simply added (ref. [32]), so that their contributions to the Wigner scattering time delay are also additive."

Generally the article is quite long and at times perhaps a bit too technical for the general audience of Nature Communications. The authors could consider placing certain parts within the supplementary information, e.g. "Behaviour of PDA curves in a Resonance", or parts of the derivation of the Fano Line-Shape analysis.

We shortened the manuscript slightly by paraphrasing several sections in a more compact manner without shortening its content. We also improved the structure of and the references to our Supplementary Notes in order to improve the readability of the manuscript. In the Supplementary Information we removed the note on the PDA for the perpendicular transition and the one on the net phase change of the PDA in a resonance (Notes 8 and 10 in the previously submitted manuscripts), since its content is not necessarily relevant for the understanding of the main manuscript and we

acknowledge the opinion of all referees that the manuscript and Supplementary Information is rather on the long side.

Small notes/typos: PDA on first use undefined (p2, 116)

PDA is now defined on its first use on page 1, line 51

Fig. 3b/d the colour scale of the line is unnecessary and makes the two traces harder to distinguish. Perhaps just use different coloured lines here.

As suggested, we modified Fig. 3b/d and show now the two different emission angles in a single color and with two different line thicknesses only.

Reference to Eq. 24/25 should presumably refer to references 24/25 (p7, 148)

The referee is right, the references were meant to point to Eqs. (8) and (9). The manuscript was corrected accordingly (page 7, line 46).

REVIEWERS' COMMENTS

Reviewer #1 (Remarks to the Author):

I appreciate the authors' response to my criticisms and those of the other referees. As I said in my previous report, I never questioned the validity of the obtained experimental and theoretical results. I think that the complementarity of the present approach with time-resolved measurements is now made clearer. Therefore I am inclined to consider the present work as suitable for publication on Nature Communications.

Reviewer #2 (Remarks to the Author):

The authors did not really respond to my report in any constructive way beyond expressing their general disagreement with my opinion and the form it was expressed in. Nothing in their response letter convinces me to change my mind or reconsider at least some of my criticisms. Nor was it, clearly, the authors' goal - their aim was to discredit my review and, convince the editors and, perhaps, the other reviewers to disregard it. That is not for me to judge if they were successful in that. I can just honestly state my own opinion, or rather restate the opinion expressed in my original report and the negative recommendation which goes with it.

Reviewer #3 (Remarks to the Author):

The authors have adequately addressed all points raised in my previous review.

I continue to be excited about this new experimental approach to extract photoemission delays for molecular systems and recommend publication of this manuscript.

Reviewer #1:

I appreciate the authors' response to my criticisms and those of the other referees. As I said in my previous report, I never questioned the validity of the obtained experimental and theoretical results. I think that the complementarity of the present approach with time-resolved measurements is now made clearer. Therefore I am inclined to consider the present work as suitable for publication on Nature Communications.

We would like to thank Referee 1 for the evaluation of our work. Thanks to the constructive criticism in his/her first report we have the feeling that we could improve our manuscript and pass our message with more clarity.

Reviewer #2 (Remarks to the Author):

The authors did not really respond to my report in any constructive way beyond expressing their general disagreement with my opinion and the form it was expressed in. Nothing in their response letter convinces me to change my mind or reconsider at least some of my criticisms. Nor was it, clearly, the authors' goal - their aim was to discredit my review and, convince the editors and, perhaps, the other reviewers to disregard it. That is not for me to judge if they were successful in that. I can just honestly state my own opinion, or rather restate the opinion expressed in my original report and the negative recommendation which goes with it.

We acknowledge the second referee's opinion on our manuscript. We would like to emphasize that we addressed all of the points that were raised in his/her first report in our response letter, and we were hoping to obtain in turn a response in order to better understand where his/her opinion stems from.

Reviewer #3 (Remarks to the Author):

The authors have adequately addressed all points raised in my previous review. I continue to be excited about this new experimental approach to extract photoemission delays for molecular systems and recommend publication of this manuscript.

We again thank Referee 3 for his/her feedback on our manuscript.